# Training Semi-Supervised Deep Learning Models with Heuristic Early Stopping Rules

## Abstract

Semi-supervised learning (SSL), especially when combined with deep learning (DL) models, is a useful technique when there is a substantial amount of unlabeled data. This is particularly relevant in healthcare applications, such as mHealth, where data is often collected through smartphones. Labels are typically obtained via self-reported questions delivered by the device and tend to have a high rate of non-response i.e., missing labels. Despite its benefit, there is a lack of objective methodology on how to train semi-supervised deep learning (SSDL) models. In this study, we propose a framework for early-stopping in SSDL that terminates learning to prevent overfitting and before the performance starts to deteriorate. Our approach focuses on three aspects: model stability, generalizability, and high-confidence pseudo-label (i.e., label assigned to unlabeled data during SSL). We first monitor changes in learned weights of the model to assess convergence, using weight stabilization. We also track cross-entropy loss, identifying which iteration of the SSL algorithm minimizes validation loss and improves generalizability. Lastly, we use a sliding window method to assess our confidence in the pseudo-labels, retaining only the most reliable labels during training. Combining these criteria, this SSDL framework can be used to train deep learning models in the context of SSL with an objective criteria that prevents overfitting and improves generalizability. We apply this SSDL training strategy to mHealth data (device sensor data and self-reported data) collected from participants in a clinical trial, which consists of 4,700 observations, 62% of which are unlabeled. Using this objective early stopping criteria for training, we achieve improvements in accuracy and F1 scores, compared to the benchmark model where the early stopping criteria is not applied.

## 1    Introduction

### 1.1    From Supervised to Semi-Supervised Deep Learning

The early stage of deep learning (DL) models primarily focused on supervised learning, used in the case where fully-labeled data are accessible in the training step. For example, supervised DL models achieved success in detecting diabetic retinopathy from retinal images (Gulshan et al., 2016) as accurately as human specialists. In another example, supervised DL has been also applied to predict cardiovascular risk from echocardiogram video data (Ouyang et al., 2020), which provided more accurate and real-time assessments so that it could fully support clinical decision-making.

There are many instances in healthcare applications where fully-labeled data is unavailable, e.g., only highly trained experts can manually annotate symptoms or diagnostic labels for medical images. Another example of unlabeled data in healthcare applications is in mHealth studies (involving smartphones) which is the context of this work. Prediction models with mHealth data often have prediction targets as user self-reports, also known as Ecological Momentary Assessments or EMA, which suffer from non-response from participants (Stone et al., 2023). Such non-responses in EMAs can be considered as unlabeled data.

Semi-supervised deep learning (SSDL) is an alternative, to include unlabeled data in the prediction model as excluding them can lead to a biased and less generalizable prediction model. For example, Bai et al. (2017) and Yu et al. (2019) demonstrated better performances with image segmentation of

cardiac MRI and 3D left atrium data, respectively. In electronic health record (EHR) analysis, Miotto et al. (2018) increased the performance of predicting a large volume of patients' health outcomes such as the onset of diseases in long-term tracking.

One significant remaining challenge in SSDL is is the lack of an objective methodology on how to train such models and when to stop training. While stopping rules have been proposed for DL models, it remains unclear how they should be implemented in semi-supervised learning (SSL) which iterates on predicting initial labels for unlabeled data (pseudo-labeling) and updating the pseudo-labels through pre-training and fine-tuning (Li et al., 2019). Thus, errors in predicted pseudo-labels from earlier iterations can accumulate and propagate through subsequent training steps—a phenomenon known as error propagation.. This propagation undermines the effectiveness of feature extraction by deep networks, which in turn impacts the model's final classification performance (Wu & Prasad, 2017; Arazo et al., 2020; Nishi et al., 2021). Furthermore, SSDL is known to tend to be overfitting on small initial pseudo-labels so that robust training strategy is demanding.

## 1.2 CONTRIBUTION

We propose three heuristic criteria, each serving as an individual early stopping rule for training an SSDL, with the objective of preventing overfitting and error propagation. By incorporating various metrics of model performance and convergence in the training phase, such as, stabilization of the weights of the DL model, criterion to improve generalizability and approach on the confidence of the pseudo-labels of unlabeled data. This is distinct from existing methods that only rely on a single metric, such as accuracy, loss function, or mean squared error (MSE) (Yalniz et al., 2019; Ouali et al., 2020).

We demonstrate the application of the heuristic stopping criteria in the SSDL framework and illustrate how each criterion works under various scenarios. Rather than relying solely on regularization or uniform criteria for model training (Sohn et al., 2020; Tarvainen & Valpola, 2017), this approach improves performance by tailoring the stopping point to specific training conditions. This scheme can be used regardless of the complexity of the task, confidence of pseudo-labels, domain-specificity of the data.

## 2 METHODOLOGY

### 2.1 DATA OVERVIEW

The data utilized in this study was collected by our research center[1]. The participants of this study undergo psychotherapy for depression for 9 weeks and are given a smartphone to monitor their behavior and augment the psychotherapy. The dataset consists of two types of measures: active and passive data.

First, active data includes patients' daily self-reported responses to instruments such as the Photographic Affect Meter (PAM, Pollak et al. (2011)), the Patient Health Questionnaire (PHQ), and homework compliance. Homework refers to psychotherapy-related tasks, which are central to Cognitive Behavioral Therapy (CBT) (Hofmann et al., 2012). These structured tasks, discussed during therapy sessions, are to be completed between sessions. Predicting non-compliance with these tasks allows for targeted interventions. For compliant participant, encouraging nudges can be sent, while non-compliant participant may receive reminders or additional support to improve task completion.

Second, passive sensing data are multivariate streams of data recorded via the sensors of the mHealth device. Some examples are step counts, travel distance from GPS locations, sleep patterns (e.g., sleep duration percentage of REM sleep hours, etc.), and time spent in human conversation (inferred from microphone). These passive observations are aggregated as daily measures so that align with the frequency of the active data.

In addition to these active and passive data, we incorporate demographic information of the participant for supporting contextual analysis. The dataset consists of 4,700 observations (43 participants

---

[1]We exclude the name of the research center, citation of the project description, and the corresponding grant number for observing anonymity.

followed up for 99 days on average), with the target variable 'homework' defined as a binary outcome: '1' representing task compliance and '0' indicating non-compliance on a daily basis. A significant portion, 62%, of the data remains unlabeled, meaning that no value has been assigned to the 'homework' variable for these entries. This high level of missing labels reflects a common issue in EMA studies, where participants may not consistently report their compliance of assigned tasks.

## 2.2 DATA PREPROCESSING

The passive and active data from mHealth devices need significant preprocessing as device use status (i.e., carrying the smartphone on person or wearing the wearable) is unknown which can result in bias in the prediction model. To this end, we conduct preprocessing in two steps. First, we use 2SpamH (Zhang et al., 2024), a preprocessing algorithm for passively sensed mHealth data, to estimate days when device use is low, which are considered as missing values; this step addresses the bias due to device non-use inherent in passive sensing data from mobile devices (Zhang et al., 2023). Second, we use the R package `missForest` (Stekhoven & Bühlmann, 2012) to impute the missing data, with estimated values derived from neighboring information (i.e., similar days with high device use). This completes our preprocessing and produces a cleaned and imputed dataset where the prediction target, i.e., homework compliance, is measured for 4,700 person-days. Next, we construct features from passive and active data for each person-day of homework compliance by constructing a 7-day and 2-day look-back window for a particular person to capture both long-term and short-term trends. During these look-back windows, we construct features that include the mean, standard deviation, and trends for each feature. All features are then standardized.

We perform cross-validation (CV) in a longitudinal participant-specific manner, such that we learn a participant's behavior through passive and active sensing for the first 21 days and provide personalized predictions after 21 days to the end of the study, which is typically 9 weeks or 63 days. This setup mirrors a real-world use case, where digital interventions are pushed based on predicted homework compliance. The first 7 days (out of 9 weeks) of the participant's homework compliance data (target of prediction) is not utilized as it is used to engineer features for the prediction model starting at day 8. For the CV, the data is randomly divided into 4-folds at participant level, i.e., each fold has approximately 10 participants and their entire active and passive data collected over 9 weeks. Then, the longitudinal data of each participant is divided into two blocks, the first block from days 0 to 21 (or 3 weeks) is retained only for training and the second block from days 22 - 63 is used for test data in the cross-validation. For each fold in the CV (e.g., Fold 1), the model is trained using Block 1 data from that fold and Block 1 and 2 data from the remaining folds. Block 2 from Fold 1 is used as the test data. This process is repeated for all folds, ensuring that each fold's Block 2 data serves as the test set once. Figure 1 provides a visual representation of how the dataset is split into folds and blocks for the CV process.

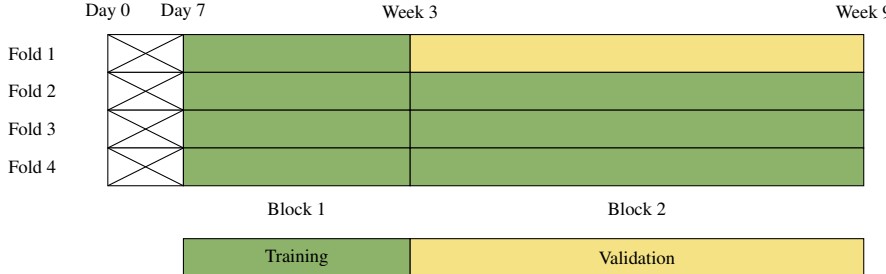

Figure 1: Visualization of Data Splitting for Subject-Dependent Cross-Validation for Fold 1

## 2.3 MODEL ARCHITECTURE

The model architecture of the proposed SSDL model is detailed in the Appendix (Figure A1). Each row in the dataset consists of 187 variables used as predictors for predicting a participant's homework compliance status on a specific day. The input layer of the model consists of 187 units, with each unit corresponding to one of these variables. Dropout with a rate of 0.3 is applied across the input layer, as well as the first and second hidden layers, to randomly deactivate 30% of the neurons

during training (i.e., their corresponding weights are not updated). This prevents overfitting by ensuring that the model does not become overly reliant on any single feature (Srivastava et al., 2014). In a neural network (NN) model, the weights are scalars representing the strength of connections between neurons across different layers. These weights are updated after each epoch using the Adam optimizer (Kingma & Ba, 2017), in our context, which adapts the learning rate for each parameter.

The first hidden layer has 25 neurons and uses the hyperbolic tangent (Tanh) activation function. The second hidden layer consists of 13 neurons and uses the the Rectified Linear Unit (ReLU) activation function. The number of neurons in the first and second hidden layers was selected through hyperparameter tuning using Bayesian optimization via KerasTuner in Python (O'Malley et al., 2019) to optimize the model's performance while maintaining an appropriate level of complexity.

Finally, the output layer consists of a single neuron with a sigmoid activation function, which produces a probability between 0 and 1 to predict the likelihood of a binary outcome. By applying dropout throughout the model and tuning the number of neurons in each layer, the architecture is designed to prevent overfitting.

## 2.4 MODEL TRAINING

Building upon the SSDL framework proposed in Li et al. (2019), we developed an heuristic approach that is tailored to our particular dataset and research objectives. In our dataset, labeled data specifically refers to observations where the homework compliance status (completion or non-completion) is known, and unlabeled data refers to those where this information is unknown. Table 1 defines all notations used throughout this paper; $i$ and $j$ are used as indices for SSL-iteration (lines 5-14 in 1) and subject. Note that $j$ is used generically; we do not define a range of values for $j$ as it varies depending on the data in question. For example, $\hat{y}_{i,j}$ is the pseudo-label of the target variable homework $y_{i,j} \in \{0, 1\}$ for participant $j$ in $\mathcal{L}_{\text{pseudo},i}$, but the range of $j$ varies based on the number of pseudo-labels assigned during SSL-iteration $i$. We ignore subscripts $i$ and $j$ in Algorithm 1 for simplicity.

| NOTATION | DESCRIPTION |
|:---:|:---|
| $\mathcal{L}$ | Labeled training dataset |
| $\mathcal{L}_{\mathcal{T}}$ | Labeled test dataset |
| $\mathcal{U}$ | Unlabeled dataset |
| $\mathcal{L}_{\text{pseudo},i}$ | Pseudo-labeled dataset of $i^{th}$ SSL-iteration |
| $M_{\text{init}}$ | Initial model (untrained NN) |
| $S$ | Number of SSDL iterations |
| $\hat{y}_{i,j}$ | Pseudo-label for participant $j$ in $\mathcal{L}_{\text{pseudo},i}$ |
| $\theta$ | Minimum probability for high-confidence pseudo-labels |

Table 1: Notations (We ignore subscripts $i$ and $j$ in Algorithm 1 for simplicity).

The algorithm begins by training the initial model (see Figure A1 for the architecture of model) on the labeled dataset to obtain a base model (line 3, Algorithm 1). This training process is conducted over 100 epochs. An epoch refers to a complete pass through the training dataset, during which the model processes every sample once and adjusts its parameters based on the loss and optimization strategy (Goodfellow, 2016).

We use this base model to generate predictions for $\mathcal{U}$ (line 4, Algorithm 1), i.e., the predicted probability of homework compliance in the unlabeled and designate observations whose $Pr(\hat{y}_{i,j}) > \theta$ as pseudo-labels in $\mathcal{L}_{\text{pseudo},i}$ (line 6, Algorithm 1). We freeze the first layer, meaning that its weights $W_{i,e}^{(1)}$ (see Appendix A.2) are excluded from the gradient updates and remain unchanged during training, while we utilize the pseudo-labels $\hat{y}_{i,j}$ to train the last two layers, the second hidden layer and output layer (lines 7-8, Algorithm 1). This heuristic approach is motivated from pre-training.

We then unfreeze the first hidden layer and fine-tune the model for another 100 epochs using the initial labeled dataset (lines 9-10, Algorithm 1). This step uses true labels to refine the model, reducing the risk of overfitting to potentially noisy pseudo-labels generated during pre-training. At each SSL iteration, we save the model's weights, biases, and optimizer state (lines 11, Algorithm 1), allowing for the retrieval of models from specific iterations. By completing all iterations and

---

**Algorithm 1** Semi-supervised Deep Learning Algorithm

---

1: **procedure** SSDL($\mathcal{L}$, $\mathcal{L}_\mathcal{T}$, $\mathcal{U}$, $M_{init}$, $S$, $\theta$)
2:      $M \leftarrow M_{\text{init}}$
3:      $M$.fit($\mathcal{L}$)                                     ▷ Base Model
4:      $\hat{y} \leftarrow M$.predict($\mathcal{U}$)
5:      **for** $i = 1$ to $S$ **do**                       // SSL-iterations
6:          $\mathcal{L}_{\text{pseudo}} \leftarrow \{(\mathcal{U}, \hat{y}) \mid \hat{y} > \theta\}$              ▷ Pseudo-labeling
7:          Set the last two layers of $M$ to trainable
8:          $M$.fit($\mathcal{L}_{\text{pseudo}}$)                          ▷ Pre-Training
9:          Set all layers of $M$ to trainable
10:         $M$.fit($\mathcal{L}$)                                 ▷ Fine-Tuning
11:         Save the current state of $M$ as $M_i$
12:         $\hat{y} \leftarrow M$.predict($\mathcal{U}$)
13:         Evaluate model $M$ on $\mathcal{L}_\mathcal{T}$ and compute performance metrics
14:      **end for**
15:      **return** $M_1, M_2, \ldots, M_S$             // Models from each iteration
16: **end procedure**

---

retrospectively evaluating performance, we avoid premature stopping. While early stopping offers computational efficiency, this approach provides a more comprehensive assessment of the model's learning trajectory. The fine-tuned model is then used to regenerate pseudo-labels for the entire unlabeled dataset (line 12, Algorithm 1). The performance metrics of interest are obtained from a separate labeled test dataset $\mathcal{L}_\mathcal{T}$ (line 13, Algorithm 1). This SSDL step repeats for a predetermined number of iterations $S$, and iterations from here on will refer to the iteration of the SSL loop.

## 2.5 CRITERIA FOR IDENTIFYING AN EARLY STOPPING RULE

In the SSDL training process described in lines 11 to 20 of Algorithm 1, determining the appropriate stopping point is critical for preventing overfitting and improving generalizability to unseen data. The following section outlines heuristic criteria developed for the early stopping rule. Each of these criteria—model stability, generalizability, and the use of high-confidence pseudo-labels—can be applied individually or combined to select the most appropriate model.

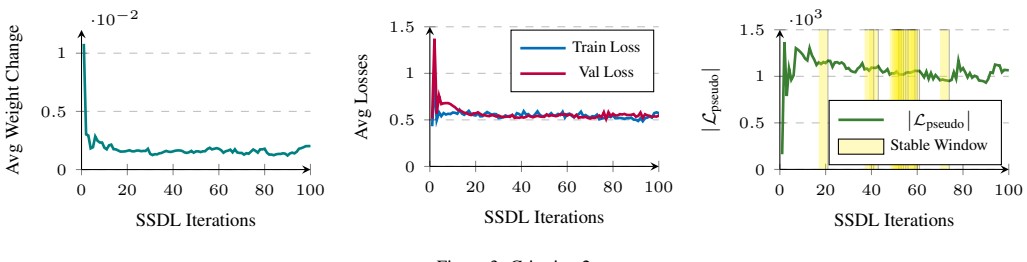

Figure 2: Criterion 1             Figure 3: Criterion 2             Figure 4: Criterion 3

### 2.5.1 CRITERION 1: MODEL CONVERGENCE

We consider the convergence of weights as an indicator that the loss function has likely reached a (local or global) minimum and that further training is unlikely to improve performance. We evaluate the convergence of the model by tracking the changes in layer-wise weights across consecutive epochs during SSL iterations. We follow the steps below to measure the model convergence:

1. Calculate the Euclidean norm of the changes in the layer-wise weights between two consecutive epochs $e$ and $e - 1$ for layer $l$ during semi-supervised iteration $i$:

$$\|\Delta W_{i,e}^{(l)}\|_2 = \|W_{i,e}^{(l)} - W_{i,e_{-1}}^{(l)}\|_2 \tag{1}$$

2. Find the maximum weight change for each layer $l$ at iteration $i$:

$$\text{MaxChange}_i^{(l)} = \max_e \|\Delta W_{i,e}^{(l)}\|_2 \tag{2}$$

3. Average these maximum weight changes across all $L$ layers for each iteration $i$:

$$\text{AvgMaxChange}_i = \frac{1}{L} \sum_{l=1}^{L} \text{MaxChange}_i^{(l)} \tag{3}$$

Figure 2 depicts the average maximum weight change during each SSL iteration. Initially, the model undergoes significant transformations as it actively acquires knowledge from its inputs. As training continue, these changes gradually decrease and stabilize. Further training brings negligible weight updates as the model becomes less sensitive to additional training data. We determine the number of training iterations for convergence using a retrospective early stopping rule, identifying the point when the average maximum weight changes reach a minimum.

### 2.5.2 CRITERION 2: GENERALIZABILITY

Generalizability refers to a model's ability to perform well on unseen data, particularly in settings where the statistical properties may vary (Gohil et al., 2024). To assess generalizability, we apply the cross-entropy loss, which is a measure of the difference between the predicted probabilities and the observed outcomes to various test datasets. The cross-entropy loss is:

$$\text{Loss}_i = -\frac{1}{n} \sum_{j=1}^{n} [y_{i,j} \log(\hat{p}_{i,j}) + (1 - y_{i,j}) \log(1 - \hat{p}_{i,j})] \tag{4}$$

where $n$ is the total number of participants in the training or validation set in question, $y_{i,j}$ and $\hat{p}_{i,j}$ are the true label and the predicted probability for participant $j$ during SSL-iteration $i$, respectively.

Figure 3 displays the cross-entropy loss values of the training set and validation set across iterations. Both losses initially decrease rapidly, indicating effective learning and sustained improvement. As training progresses, these losses gradually stabilize and converge to each other. The criterion for determining the optimal number of iterations is the point at which the validation loss is minimized.

### 2.5.3 CRITERION 3: CONFIDENCE OF PSEUDO-LABELS

In SSDL, the base model trained on labeled data generates the initial pseudo-labels for unlabeled data and incorporates these pseudo-labels into the model training process. After each $i^{th}$ SSL iteration, the updated model $M_i$ trained both on labeled data and pseudo-labels from the previous iteration generates new pseudo-labels. Hence, the level of confidence we have in these pseudo-labels is crucial. Low-confidence pseudo-labels can introduce noise into the training process and reduce model performance (Wu & Prasad, 2017). We implement the following steps to monitor the number of high-confidence pseudo-labels.

1. Retrieve the count of high-confidence pseudo labels from $\mathcal{L}_{\text{pseudo},i} = \{\hat{y}_{i,1}, \hat{y}_{i,2}, \ldots\}$ for each iteration $i = 1, \ldots, 100$. High-confidence pseudo-labels refer to those predicted labels for which the model's predicted probability exceeds $\theta$ i.e., $Pr(\hat{y}_{i,j}) > \theta$. Note that $|\mathcal{L}_{\text{pseudo},i}| \geq 2$ since we require that there is at least one pseudo-label of each class (compliance and non-compliance).

2. For each SSL iteration $i$ , we consider a sliding window over $w$ iterations with step size of 1, i.e., $i$ to $i + w - 1$, and compute the average count of the number of high-confidence pseudo-labels:

$$\mu_i = \frac{1}{w} \sum_{k=i}^{i+w-1} |\mathcal{L}_{\text{pseudo},k}| \tag{5}$$

, where $|\mathcal{L}_{\text{pseudo},k}| = \sum_j \text{I}[Pr(\hat{y}_{k,j} > \theta)]$.

3. With the same sliding window, calculate the standard deviation of the counts:

$$\sigma_i = \sqrt{\frac{1}{w} \sum_{k=i}^{i+w-1} (|\mathcal{L}_{\text{pseudo},k}| - \mu_k)^2} \tag{6}$$

4. Identify the stable window where the difference in the rolling average between consecutive windows (Equation 5) is less than the average threshold $\epsilon_\mu$, and the standard deviation within the current window (Equation 6) is below the standard deviation threshold $\epsilon_\sigma$:

$$\text{StableWindow} = \{i \in \mathbb{N} \cap [1, 100 - w + 1] \mid |\mu_i - \mu_{i-1}| \le \epsilon_\mu, \ \sigma_i \le \epsilon_\sigma\} \quad (7)$$

Figure 4 illustrates the number of high-confidence pseudo-labels generated during each SSL iteration. Initially, the chart displays fluctuations, reflecting the model's active learning phase as it explores and refines category boundaries. Over iterations, the count stabilizes, and multiple stable windows are observed. We select the iteration with the highest rolling average of high-confidence pseudo-labels, in order to identify the set of pseudo-labels that are most stable over SSL-iterations.

### 2.5.4 COMBINED CRITERIA

We determine an appropriate point to halt the training of our SSDL model by integrating three key criteria: model convergence, generalizability, and high-confidence of pseudo-labels. To select a suitable iteration, we follow a process that integrates these criteria in tandem.

First, we select the top $s(< S)$ candidate iterations based on each specific criterion: model stability, generalizability, and high-confidence pseudo-labels. These $s$ candidates are selected from the $S$ SSL iterations, where $S$ represents the total number of training iterations. Here, $s$ can be chosen to ensure enough iterations are considered for evaluation. Once we have the top $s$ iterations for each criterion, we consider the overlap between them. This ensures that the iterations we focus on achieve balanced performance across multiple metrics. For the overlapping iterations, we compute an average rank based on how well each iteration performs in each criterion. Instead of using the raw performance values, we rank the iterations for each criterion. For instance, if a particular iteration ranks $10^{th}$ in model convergence criterion, $12^{th}$ in generalizability, and $2^{nd}$ in pseudo-label quality, its average rank would be the mean of these rankings: $\frac{10+12+2}{3} = 8$. This approach allows us to aggregate performance across criteria; weighted mean can also be used when favoring specific metrics over others. Finally, we select the iteration with the lowest average rank, prioritizing iterations that perform well across all criteria.

## 3 RESULTS

Table 2 presents the model's performance across various evaluation metrics for each criterion applied during training. Cells are color-coded in shades of green, with darker shades indicating higher scores. The Benchmark column reflects results without applying any criterion for comparison. Definitions of all metrics used in this analysis are provided in Appendix A.3.

|  | Benchmark | Model Convergence | Generalizability | Confidence of Pseudo-Labels | Combined Criteria |
|---|---|---|---|---|---|
| **Sensitivity** | 0.7559 | 0.7883 | 0.7497 | 0.7576 | 0.7823 |
| **Specificity** | 0.7955 | 0.7723 | 0.8119 | 0.8007 | 0.7816 |
| **PPV** | 0.9024 | 0.8965 | 0.9101 | 0.9043 | 0.9011 |
| **NPV** | 0.5821 | 0.6088 | 0.5794 | 0.5867 | 0.5997 |
| **AUC** | 0.8228 | 0.8274 | 0.8316 | 0.8163 | 0.8338 |
| **Accuracy** | 0.7677 | 0.7831 | 0.7684 | 0.7695 | 0.7825 |
| **F1 Score** | 0.8203 | 0.8368 | 0.8197 | 0.8218 | 0.8360 |

Table 2: Performance Metrics Across Different Criteria

The model convergence criterion demonstrates strong performance in sensitivity, negative predictive value (NPV), accuracy, and F1 score, indicating that the model is highly sensitive and precise in its predictions. These metrics achieve the top scores, though there is a trade-off with lower specificity and positive predictive value (PPV), suggesting a tendency to generate more false positives. This highlights the challenge of balancing true positive detection with minimizing false positives.

The generalizability criterion, focused on minimizing validation loss, excels in specificity and PPV, reflecting strong performance in correctly identifying both true negatives and positives. However, it

performs lower in sensitivity, NPV, accuracy, and F1 score, implying a potential risk of missing true positives, which could lead to reduced overall accuracy.

The quality of pseudo-labels criterion shows moderate performance in specificity and PPV, but underperforms in sensitivity, NPV, accuracy, and F1 score. While the model is reasonably effective at distinguishing true negatives and positives, it struggles to achieve balance across the board. This emphasizes the importance of reliable pseudo-labels in driving successful model training.

Combining all criteria results in a model results in the best AUC, high accuracy and F1 score, moderate sensitivity and NPV, and good specificity and PPV. Thus the combined-criteria approach produces a well-rounded model that maximizes performance across all metrics without major trade-offs.

## 4 DISCUSSION

Developing and optimizing our SSDL framework requires careful adjustment of various parameters to achieve the best performance. In deep learning architecture (Figure A1), parameters such as the number of layers, units per layer, activation functions, learning rate, and epochs play a crucial role. For example, model complexity, determined by layers and units, must be balanced to avoid underfitting or overfitting. This parameter tuning is compounded when a SSL framework is used with deep learning. In our SSDL framework (Algorithm 1), there are additional key parameters that include the maximum number of iterations $S$, the confidence threshold for pseudo-labels $\theta$, and the sliding window size $w$ in $5 - 7$. When proposing an early stopping rule based on a combination of three criteria—model stability, generalizability, and pseudo-label confidence—we introduce an additional parameter, $s$, which refers to the top iterations selected based on performance across these criteria (see Section 2.5.4). A small $s$ risks insufficient overlap between performance with respect to the three criteria, while a large $s$ could potentially add noise to the selection of the best performing iteration. In summary, a unified framework to effectively evaluate this complex training procedure is needed.

## 5 CONCLUSION

This study presents a framework for training a SSDL model tailored for healthcare applications, addressing the challenge of limited labeled data. By integrating heuristic criteria—model convergence, generalizability, and confidence of pseudo-label—our approach aims to enhance the robustness of the model.

Each criterion contributed uniquely to evaluating the model's performance. The model convergence criterion improved sensitivity and overall accuracy, while the generalizability criterion exhibited strengths in specificity and positive predictive value. Our results suggest that combining these criteria resulted in a more balanced and stable model, yielding high performance across multiple metrics.

While the approach outlined in this paper serves as an heuristic guide for developing SSDL models, it is by no means a definitive solution. We aim to provide a framework that encourages further innovative research in the field of SSDL, particularly as the high volume of unlabeled data is a common challenge across many real-world applications. By adapting these principles to diverse contexts, this framework has the potential to offer valuable insights across a wide range of applications.

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

# A APPENDIX

## A.1 MODEL ARCHITECTURE

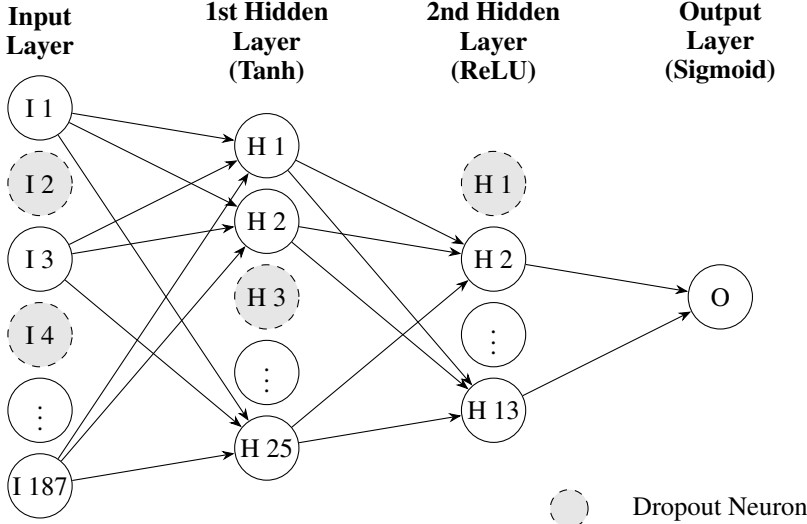

Figure A1: SSDL Model Architecture with Dropout Layer.

## A.2 LAYER-WISE WEIGHT MATRICES

- Input Layer to 1st Hidden Layer (Tanh)

$$W_{i,e}^{(1)} \in \mathbb{R}^{187 \times 25}$$

- 1st Hidden Layer (Tanh) to 2nd Hidden Layer (ReLU)

$$W_{i,e}^{(2)} \in \mathbb{R}^{25 \times 13}$$

- 2nd Hidden Layer (ReLU) to Output Layer (Sigmoid)

$$W_{i,e}^{(3)} \in \mathbb{R}^{13 \times 1}$$

## A.3 DEFINITIONS OF PERFORMANCE METRICS

- Sensitivity: The ability to correctly identify positive instances (also known as recall).
- Specificity: The ability to correctly identify negative instances.
- PPV (Positive Predictive Value, Precision): The precision or the proportion of true positives among all predicted positives.
- NPV (Negative Predictive Value): The proportion of true negatives among all predicted negatives.
- AUC (Area Under the Curve): A measure of the ability of the classifier to distinguish between classes.
- Accuracy: The overall proportion of correctly classified instances.
- F1 Score: The harmonic mean of precision and recall, which balances the trade-off between the two.

