# OpenReview forum: "Training Semi-Supervised Deep Learning Models with Heuristic Early Stopping Rules"
_ICLR.cc/2025/Conference — Submitted to ICLR 2025_

### Official Review · Reviewer_p3TY · 2024-10-26

**Soundness:** 2
**Presentation:** 3
**Contribution:** 2
**Rating:** 3
**Confidence:** 4

**Summary:**

This paper proposes a framework for early stopping in semi-supervised learning. The paper introduce three heuristic criteria for early stopping: model stability (based on weight convergence), generalizability (based on validation loss) and pseudo-label confidence (over a rolling window). They demonstrate their approach on a single dataset compared to baseline models that do not implement the early stopping criteria.

**Strengths:**

- Originality: The proposed early stopping criteria, especially the combination of weight stabilization, generalizability and pseudo-label confidence are interesting and could be used to guide SSL training

- Quality: The experiments show on the single dataset that they improve performance

- Clarity: The paper is well written and most importantly the problem is well-motivated with relevant real-world examples from healthcare.

- Significance: The method addresses a practical problem in healthcare ML applications where SSL is widely used

**Weaknesses:**

The two major weaknesses are related to experiments and baselines to be able to have findings that are truly generalizable to SSL and PL in general.

(1) Limited experiments: the paper only uses a single dataset — so it’s hard to judge if this could generalize beyond just this setting. It is recommended that the authors include additional datasets as is common in SSL papers to demonstrate how the method behaves on datasets with different properties.

(2) Limited baselines: the paper only uses a baseline without these heuristics. Two comparisons are needed:

(a) Related approaches to early stopping such as Bai et al: Understanding and Improving Early Stopping for Learning with Noisy Labels

(b) SSL baselines: The paper only studies vanilla PL. However, there have been significant advancements in the SSL SOTA and it’s important to understand if the heuristics of early stopping help these paradigms of SSL as well, which change the pseudo-label selector functions: UPS (Rizve et al., 2021), FlexMatch (Zhang et al., 2021), SLA (Tai et al., 2021), CSA (Nguyen et al., 2022).

UPS: https://arxiv.org/abs/2101.06329
Flexmatch: https://arxiv.org/abs/2110.08263
SLA: https://arxiv.org/abs/2102.08622
CSA: https://arxiv.org/abs/2206.05880

The authors are encouraged to cover these different experimental dimensions to help support their findings and whether they generalize

**Questions:**

Questions about sensitivity:
(a) How would the heuristics be affected on datasets with different proportions of labeled/unlabeled data?
(b) How are the heuristics affected by different SSL paradigms or approaches as suggested above (e.g. UPS, Vanilla PL, SLA etc)?
(c) For the combined criteria approach, how do you determine the weights for each criterion? Have you explored different weighting schemes?
(d) For the confidence heurisitic how do you select the threshold? In particular, let’s say you used different PL approaches (vanilla vs UPS), they have different selector functions to let sample through to different PL iterations. This might affect the confidence and how would then affect the heuristic

---

### Official Review · Reviewer_bf16 · 2024-10-31

**Soundness:** 2
**Presentation:** 2
**Contribution:** 1
**Rating:** 3
**Confidence:** 4

**Summary:**

Paper proposes a framework for training semi-supervised deep learning (SSDL) models with a focus on early stopping rules. The work is motivated by healthcare applications, specifically mHealth data where many observations lack labels.

**Strengths:**

1) Addresses a real-world problem in healthcare data analysis where unlabeled data is common.
2) The framework could be valuable for other domains with similar challenges.

**Weaknesses:**

1) Lacks theoretical analysis of how the criteria interact with each other as described in Section 2.5.4.
2) No theoretical justification for why these three specific criteria were chosen. The proposed criteria are not new and has been explored earlier in semi-supervised learning.
3) Only tested on a single dataset with relatively few participants. There are several other time-series datasets in medical/health domain that would be suitable for evaluation.
4) No comparison with other state-of-the-art SSDL methods. At the moment, there are no comparison with how existing semi-supervised learning methods work in this setting. It is hard to see the feasibility of the proposed method.
5) Limited ablation studies to understand the contribution of each component.
6) The combined criteria method relies on ranking, which might not scale well with dataset size and models.

**Questions:**

How sensitive are the results to the choice of hyperparameters, particularly the confidence threshold θ and the sliding window size w?
What is the impact of the dataset size and the  percentage of labeled data on the effectiveness of each criterion?
How would the framework perform on datasets from different domains or with different label sparsity patterns?

---

### Official Review · Reviewer_BGV5 · 2024-11-02

**Soundness:** 2
**Presentation:** 3
**Contribution:** 1
**Rating:** 3
**Confidence:** 4

**Summary:**

The paper propose a framework for early-stopping in training semi-supervised learning model. The proposed framework work by
1. monitor the changes in learned weights of model to assess convergence
2. track cross-entropy loss (identifying which iteration of SSL algorithm minimizes validation loss)
3. Use sliding window to assess confidence in pseudo-labels, retaining only the most reliable labels during training

**Strengths:**

This paper present an interesting application of semi-supervised learning -- mHealth data.

**Weaknesses:**

1. Lack of Novelty: The heuristics proposed—such as monitoring changes in learned weights, tracking losses, and assessing the stability/confidence of pseudo-labels—are already widely used in day-to-day model training practices by researchers. I don't see anything new.

2.  Throughout the paper, the authors incorrectly referring to properties of pseudo-labeling (PL)-based methods as characteristics of all Deep SSL methods without recognizing that PL-based method is only one type of SSL method. Other methods, like the consistency regularization based method, does not necessarily involves pseudo-labels in the training process.

3. The evaluation is also too thin compared to what is typically expected at a conference of this level

**Questions:**

NA

---

### Official Review · Reviewer_qeQh · 2024-11-03

**Soundness:** 2
**Presentation:** 2
**Contribution:** 2
**Rating:** 3
**Confidence:** 4

**Summary:**

The paper proposes a heuristic method for early stopping in semi-supervised learning, which is a combination of three criteria of early stopping. The experiment on mHealth data shows a better performance than other early stopping method.

**Strengths:**

The paper is well-motivated and try to address an important problem of early-stopping in semi-supervised learning.

**Weaknesses:**

- The experiment is conducted on only one dataset with only 4700 observations and only 1800 labeled without an error bar. The results may not be statistically significant.

- No related works section for discussing related work of SSL/early stopping.

- The early stopping criterion higher depends only on the architecture of the model and how the model is trained. For example, in Figures 2, 3, 4, it seems the training converges faster and the validation loss does not go up. However, in most cases, when training a network, validation loss will go up. The experiment may not reflect the real-world problem.

**Questions:**

See weakness.

---

### Meta-Review · Area_Chair_jizv · 2024-12-17

**Metareview:**

This paper proposes a framework for early-stopping in semi-supervised deep learning that terminates learning to prevent overfitting and before the performance starts to deteriorate. This framework combines three criteria of early stopping. This paper conducts experiments on the mHealth dataset, which shows that the proposed method can achieve better accuracy and F1 scores.

This paper receives scores of 3, 3, 3, 3, which means that all the reviewers recommend rejection. The reviewers think that the novelty of this paper is limited, the experiments are not extensive, and the results are not significant enough. There are also many other issues pointed out by the reviewers but unsolved in this paper.

The authors did not provide any rebuttal for the reviewers' comments.

Therefore, I recommend rejection.

**Additional Comments On Reviewer Discussion:**

No rebuttal is provided by the authors.

---

### Decision · Program_Chairs · 2025-01-22

Reject